# Verification of Criterion-Related Validity for Developing a Markerless Hand Tracking Device

**DOI:** 10.3390/biomimetics9070400

**Published:** 2024-07-02

**Authors:** Ryota Suwabe, Takeshi Saito, Toyohiro Hamaguchi

**Affiliations:** 1Department of Rehabilitation, Graduate School of Health Sciences, Saitama Prefectural University, Saitama 343-8540, Japan; 2681309e@spu.ac.jp (R.S.); saitotakeshi@tdc.ac.jp (T.S.); 2Department of Rehabilitation, Tokyo Dental College Ichikawa General Hospital, Chiba 272-8513, Japan

**Keywords:** MediaPipe, hand tracking, paralysis estimation

## Abstract

Physicians, physical therapists, and occupational therapists have traditionally assessed hand motor function in hemiplegic patients but often struggle to evaluate complex hand movements. To address this issue, in 2019, we developed Fahrenheit, a device and algorithm that uses infrared camera image processing to estimate hand paralysis. However, due to Fahrenheit’s dependency on specialized equipment, we conceived a simpler solution: developing a smartphone app that integrates MediaPipe. The objective of this study was to measure hand movements in stroke patients using both MediaPipe and Fahrenheit and to assess their criterion-related validity. The analysis revealed moderate-to-high correlations between the two methods. Consistent results were also observed in the peak angle and velocity comparisons across the severity stages. Because Fahrenheit determines finger recovery status based on these measures, it has the potential to transfer this function to MediaPipe. This study highlighted the potential use of MediaPipe in paralysis estimation applications.

## 1. Introduction

The human “hand” contains a unique set of features within its complex assembly of joints, finely tuned for its specialized role as a delicate manipulative organ [1]. Understanding that hand function relies on specialized areas within the central nervous system to monitor the environment via sensory receptors and control fine movements through both exogenous and endogenous muscles is crucial [2]. Therefore, impairments in finger function following stroke are significant issues that directly affect the activities of daily living (ADL) and quality of life (QOL), making the improvement of finger function a key goal in the rehabilitation of stroke survivors [3].

To rehabilitate paralyzed fingers, it is important to optimize the quality and quantity of tasks. Quality refers to the appropriate level of difficulty, and quantity refers to the appropriate amount of practice [4,5]. To improve the quality of training, it is necessary to grade training according to its severity [6], and accurate evaluation is required to make this possible.

Typical measures of motor function in stroke include the Fugl-Meyer assessment [7], Brunnstrom stage [8], and Stroke Impairment Assessment Set (SIAS) [9], all of which score motor function based on the expert’s visualization of motor onset and end. Since motor function assessment is performed and judged by experts, it can be performed in a short time without special equipment and has been widely used. However, visual evaluation of the starting and stopping positions of movements does not include factors related to the speed and trajectory of hand movements. In addition, the accurate evaluation of complex and high-speed hand motions using human vision is limited.

The development of image analysis in recent years has been remarkable, and various three-dimensional (3D) motion analyzers have been applied to provide temporal and spatial analyses of human motion. Our research team has also demonstrated that a binocular camera can be used to determine the severity of paralysis [10,11]. This device, called the Fahrenheit development code, is a method for recording hand motions and estimating hand functions by estimating joint centers and calculating motion features from different images captured by binocular cameras. Fahrenheit can help both the rehabilitation specialist and the patient record and analyze the motor functions of the patient’s hand to determine the effectiveness of training. However, Fahrenheit requires a dedicated infrared camera and computer terminal running an application for analysis, making its use at the bedside somewhat labor-intensive. To widely apply this system in clinical practice, we conceived the idea of implementing a dedicated application on a smartphone using a camera. MediaPipe [12] is an open-source AI solution that can estimate joint centers from videos of fingers captured using an optical monocular camera. In fact, accuracy comparisons between infrared and optical cameras have been made using pose discrimination accuracy [13,14]; however, because they do not consider the temporal component, they do not evaluate the equivalence of motion velocity and trajectory, which is our goal. To replace MediaPipe with the binocular camera imaging function of Fahrenheit, we must verify the equivalence of the hand motion estimation.

Therefore, in this study, we examined the temporal and spatial criterion-related validity of the Fahrenheit and MediaPipe functions. If the equivalence of both functions is ensured, a simple smartphone-based paralysis estimation application can be developed. This will not only assist physicians and therapists in their clinical evaluations but also serve as a basis for the development of independent training devices for patients.

## 2. Materials and Methods

### 2.1. Research Design

This study is a retrospective observational design.

### 2.2. Target

Patients with post-stroke sequelae who participated in the Fahrenheit Development Project at Saitama Prefectural University were included in this study. The participants were patients aged 20 years or older who were admitted to the Tokyo Dental University Ichikawa General Hospital between 1 June 2016 and 31 March 2021 and were recruited from among all inpatients who had suffered a stroke. The patients were classified into 6 groups, stages I–VI, according to the Brunnstrom Recovery Stage (BRS) criteria, with reference to previous studies [10,11]. Using G*power to construct models for F tests and ANCOVA, encompassing fixed effects, main effects, and interactions (with α = 0.05, 1 − β = 0.8, effect size f = 0.5, and six groups), the minimum sample size was calculated to be 34. The exclusion criteria were as follows: (1) history of multiple cerebral infarctions or transient ischemic attacks; (2) loss of one or more fingers; (3) severely limited range of motion (ROM); (4) difficulty in understanding verbal instructions due to impaired consciousness, dementia, or aphasia; (5) a history of stroke or transient ischemic attack; (6) patients who needed bed rest; (7) patients who had difficulty maintaining a sitting position for at least 30 min, and (8) patients who had difficulty maintaining the required position during measurements even with assistance. Data in which the patient’s fingers were not included in the angle of view from which the image was taken were excluded from the analysis.

### 2.3. Device

#### 2.3.1. MediaPipe

MepipPipe is an open-source machine learning (ML) solution framework provided by Google Inc. There are several solutions available for MediaPipe; in this study, we used MediaPipe Hands, a hand-tracking solution released in 2019. MediaPipe Hands uses machine learning to infer 21 3D hand–finger landmarks from a single frame captured by a monocular camera, providing highly accurate hand and finger tracking. According to Google AI, Blogs [15], the average accuracy of hand tracking is 95.7%. One of the features of this system is that it runs in mobile and powerful desktop environments [12].

#### 2.3.2. Fahrenheit

Fahrenheit, developed by Hamaguchi et al., (code name, Fahrenheit; patent number, 6375328; Saitama Prefectural University, Japan) contains an infrared sensor, the Leap Motion Controller (LMC), and obtains the coordinates of the joint centers using Unity game software (https://unity.com/ja) (accessed on 30 June 2024). Fahrenheit has 24 joint landmarks and can measure the ROM of the hand based on conventional joint range-of-motion measurements. Fahrenheit was operated on a laptop computer with a 64-bit Windows 8 (Microsoft, Kobe, Japan), and the generated data were stored on the hard drive of the same computer. It was implemented with a resolution of 0.001 mm and a sampling rate of 60 frames per second (fps) [10].

#### 2.3.3. Basic Performance of Fahrenheit and MediaPipe

To track delicate hand movements, including dexterity movements, in real time, the processing frame rate of hand tracking must be sufficiently fast. In terms of frame rate, Fahrenheit is superior because processing is performed on the device, whereas MediaPipe is considerably affected by the frame rate of the data from the camera used and the performance of the terminal running the processing algorithm (Table 1).

Regarding joint landmarks, Fahrenheit includes the metacarpals of each finger, whereas MediaPipe does not. Therefore, it is difficult to obtain accurate flexion angles of finger MP joints using conventional ROM measurement methods. Fahrenheit is limited to 90° in terms of the estimated range of motion of the hand MP joints, and values exceeding 90° are corrected to 90° or less.

### 2.4. Protocol

The patients were seated in a chair or wheelchair with the shoulder joint in a resting position, elbow joint in approximately 90° of flexion, forearm in a rotated position, and wrist in an intermediate position, and their hand movements were measured. The subjects were instructed to assume the starting posture of each task at approximately 20 cm above Fahrenheit and to maintain the starting posture for 5 s from the starting cue while watching the monitor on a laptop computer. After 5 s, on the verbal cue, the participants were instructed to repeat hand flexion and extension for as long as possible for 20 s and then maintain the maximum reaching position for 15 s after the start cue. Participants were provided with virtual hand images obtained from the recordings to show their hand movements in real time. The coordinate data obtained by the infrared camera were used as the Fahrenheit data (Figure 1a,b). The videos for the MediaPipe analysis were captured from a video camera used to capture the entire hand motion during the acquisition of the Fahrenheit data. The video was divided into images at a sampling rate of 30 fps, the coordinates were extracted from each image using MediaPipe keypoints, and the keypointed images were converted to video. The images with keypoints were converted into movies (Figure 1c). In both cases, data from 10 to 20 s after the start of the measurement were used for analysis.

### 2.5. Preprocessing

#### 2.5.1. Data Specification and Angle Conversion

Because both Fahrenheit and MediaPipe were taken from different angles of view, the acquired data were angle-transformed and the evaluations were aligned. In addition, considering the occlusion task of Fahrenheit, in which the DIP joint of each finger is hidden when the hand is maximally flexed, and the task of MediaPipe, in which there are no coordinate data for the metacarpals of each joint, the PIP joint of each finger was targeted for analysis. Fahrenheit’s angular transformation method was the same as that used in previous studies [10]. For Mediapipe, a function was programmed to acquire the data. This function is used to calculate the angle of each joint of the body. The angles were calculated using analytic geometry. For example, there are three points, A (*x*1, *y*1, *z*1), B (*x*2, *y*2, *z*2), and C (*x*3, *y*3, *z*3) (Figure 2). Consider A and C as a skeletal structure (two bones). The intersection of line segments A and C lies on point B. The angle between A and C can be found as follows:

Find the inner product of AB and BC:(1)$$d1=\vec{AB}\cdot \vec{BC}$$

Find the norm (distance) of AB/BC:(2)$$d2=\left\|AB\right\|\cdot \left\|BC\right\|$$

Convert the cosine of the angle formed by $\vec{AB}\cdot \vec{BC}$:(3)$$\mathrm{angle}=\mathrm{arc}\;\cos\;(\frac{d1}{d2})\;(\mathrm{At}\;\mathrm{this}\;\mathrm{time},-1<\frac{d1}{d2}<1)$$

#### 2.5.2. Smoothing Process

A low-pass filter (Butterworth 2nd order, cutoff 12 Hz) was used for the first processing of the angle-transformed data. This method is commonly used for marker-based motion capture data, and the optimal cutoff values have been reported [16]. Second, optimal ensemble Kalman smoothing was performed [17]. The ensemble Kalman filter involves a two-way pass to determine the optimal state estimation for a given keypoint trajectory. Hyperparameters, including measurement and transition noises, were optimized using a grid search and cross-validation (Figure 3).

#### 2.5.3. Detection and Completion of Misestimated Frames

Based on previous studies, misestimation of the coordinate data at frame i was detected by comparing the absolute value of the difference between the previous and subsequent frames [18]. The absolute value of the difference is defined as:(4)$$d_{x}=(\left|x_{2}-x_{1}\right|,\ldots ,\left|x_{i}-x_{i-1}\right|,\ldots )$$

The mean value and standard deviation were calculated for dx, and the acquired xi values were considered as misestimated frames and excluded if any value exceeding the mean ± 3SD was detected. Each frame was then linearly interpolated, and spline smoothing was subsequently applied to obtain new data (Figure 4).

### 2.6. Statistical Analysis

#### 2.6.1. DTW Distance

For the preprocessed data, the maximum flexion angle, maximum angular velocity, and average angular velocity were compared for each BRS phase. The DTW distance was calculated for each subject, and the degree of agreement was calculated using cross-correlation analysis.

DTW is a method for calculating the similarity between two time series data with different phases [19]. DTW calculates the distance between each element of two time series data on a round-robin basis and then maps the two time series data to minimize the total distance (DTW distance) between the two time series data. Even if the lengths of the two time series data are different, a DTW correspondence is possible. The DTW distance *D*(*A*, *B*) of two time series data, *A* = {*a*1, *a*2, …, *aM*} and *B* = {*b*1, *b*2, …, *bN*}, is defined as follows:*D* (*A*, *B*) = *d* (*M*, *N*)(5)

Let *C* be the distance function between two time series data and *C*(*am*) be the distance between element *bn* of one time series data and element *bn* of the other time series data, then *d* (*m*, *n*) is calculated recursively as follows:(6)$$d\;(m,n)=C(am,bn)+min\left\{\begin{array}{c} d(m-1,n)\\ d(m,n-1)\\ d(m-1,n-1) \end{array}\right\}$$

The conditional equation for this case is as follows:*d* (0, 0) = 0, *d* (*m*, 0) = *d* (0, *n*) = ∞
1 ≤ *m* ≤ *M*, 1 ≤ *n* ≤ *N*, *m*, *n* ∈ ℕ (2.4)(7)

A drawback of DTW is its vulnerability to noise. When noise occurs, the actual distance between elements that should correspond to each other may differ from the distance between elements that should correspond to each other, resulting in inaccurate correspondence. The Euclidean distance between two elements is often used as the distance function of DTW. In this case, DTW is robust to the expansion and contraction or shift of time series data in the time direction but weak against the expansion and contraction or shift in the space direction. In other words, when two time series data have the same trajectory but move in the same spatial direction, accurate correspondence is not possible [19]. An example of mapping using DTW is shown in Figure 5.

#### 2.6.2. Cross-Correlation Analysis

Using Lin’s cross-correlation coefficients (CCC) [20], we evaluated the criterion-related validity of Mediapipe and Fahrenheit. Fleiss proposed CCC limits that were used for agreement interpretation: very good (CCC > 0.9), acceptable (0.71 < CCC0.9), moderate (0.51 < CCC < 0.7), poor (0.31 < CCC < 0.5), or no agreement (CCC < 0.31) [21].

## 3. Results

### 3.1. Subjects

Patients were classified into six groups according to disease severity per the BRS: stage I (2 patients), stage II (8 patients), stage III (6 patients), stage IV (6 patients), stage V (9 patients), and stage VI (9 patients) (Table 2).

### 3.2. Comparison of Corrections

False estimation frames were detected in three patients using MediaPipe and in nine patients using Fahrenheit. The variation in values in patients with BRS I and II without joint motion was considered noise, and the amount of noise for each patient was evaluated using the Mean Absolute Deviation (MAD). The average values for MediaPipe and Fahrenheit were 2.46 and 1.42, respectively. After smoothing and correction, the MAD improved to an average of 0.81 for MediaPipe and 0.02 for Fahrenheit (Table 3).

### 3.3. Comparison of the Results

The three therapists agreed on the results of the BRS for all subjects on the day of the experiment. The histograms of the middle finger PIP recorded during the 10 s exercise period for each BRS are shown in Figure 6. The histograms show that the percentages of extension (approximately 0–20°) and flexion (approximately 80–100°) increased as the BRS increased, and the distribution became closer to bimodal. This trend was common for both MediaPipe and Fahrenheit (Figure 6). 

The maximum flexion angle of each finger increased with the BRS stage (Spearman’s rank correlation coefficient; MediaPipe: r = 0.723, *p* = 0.002; Fahrenheit: r = 0.691, *p* = 0.04) (Table 4). The maximum angular velocity of each finger increased during the BRS phase (Spearman’s rank correlation coefficient; MediaPipe: r = 0.784, *p* = 0.0001; Fahrenheit: r = 0.796, *p* = 0.001) (Table 5). The mean angular velocity of each finger increased during the BRS phase (Spearman’s rank correlation coefficient; MediaPipe: r = 0.816, *p* < 0.0001; Fahrenheit: r = 0.832, *p* < 0.001) (Table 5).

### 3.4. Comparison of Agreement between Measurements

The DTW distance between MediaPipe and Fahrenheit for each subject and the results of the cross-correlation analysis are shown as box plots for each BRS (Figure 7). The results showed a “moderate” or “better” correlation for all data, except for patients with BRS I and II (DTW = 0.092–0.579, CCC = 0.374–0.986). In particular, the middle and ring fingers exhibited a strong average correlation (DTW mean = 0.216, CCC = 0.758). The data of the 10 patients with BRS I and II were highly variable (DTW = 0.106–0.535, CCC = 0. 251–0.963), and two patients showed a weak correlation (CCC < 0.3).

## 4. Discussion

This study verified the criterion-related validity of MediaPipe and Fahrenheit. For the data preprocessed by angle transformation and smoothing, the CCC results and DTW distance showed more than moderate correlations. The results for the peak angle and peak angular velocity were also in general agreement. In a previous study on the development of Fahrenheit, the peak angle and peak angular velocity indicated by the LMC were evaluated to distinguish the recovery state of the finger, and identifiers were created [10]. These identifiers were eventually developed into Fahrenheit as an application for evaluating the recovery stage of paralysis. Based on the concordance of the measurements shown in this study, we believe that the transfer of the Fahrenheit functionality to MediaPipe, which can be used on smartphones and tablet devices, will enhance the effectiveness of hand-movement analysis in clinical practice. This motion analysis system integrated with a camera for capturing video images can analyze minute changes and movement patterns of the subject and will be a powerful tool for sharing practice goals between therapists and patients in evaluation and treatment.

Another advantage of moving from Fahrenheit to MediaPipe is that MediaPipe is a simple motion analysis system. In general, image processing methods, such as MediaPipe, have a wider effective distance for hand tracking than detection algorithms based on LMC [12,14]. Fahrenheit’s data acquisition was limited by the installation position, subject’s posture, and calibration [11]. The MediaPipe data were obtained from a video camera that was used to capture the hand motions from the start to the end of the hand motions during the Fahrenheit data acquisition. However, the position of the camera and the ratio of the hand on the image to that on the entire screen were not specified. The reason why the agreement of the index and little fingers was lower than that of the other fingers is thought to be that the index and little fingers were occluded by the other fingers and did not appear on the screen sufficiently. Even under these conditions, moderate or better agreement suggests the usefulness of MediaPipe.

In the analysis of the time series data, some limitations of both Mediapipe and Fahrenheit were identified. One of the limitations of Mediapipe is its high noise levels. Because both Mediapipe and Fahrenheit use frame processing to detect hand motions, this limitation may be directly related to the high accuracy of MediaPipe. Although patients with BRS I and II are inherently incapable of voluntary hand movements, angle fluctuations were detected in both MediaPipe and Fahrenheit. The MAD of variation was 2.46 for MediaPipe and 1.42 for Fahrenheit. In other words, the raw data for MediaPipe showed variability within 2.4° on average of the mean value. After preprocessing, based on a previous study, the MAD improved to an average of 0.81 for MediaPipe and 0.02 for Fahrenheit. Fahrenheit eliminated most angle fluctuations, whereas those of MediaPipe remained. These factors can be attributed to problems with the detection algorithm and its accuracy. The frame processing in both MediaPipe and Fahrenheit does not retain the time axis information of the previous and subsequent frames; therefore, the detected hand position deviates from the previous and subsequent frames, which results in noise. Previous studies have also shown that accuracy can be improved by improving the ratio of the hand to the entire screen and resolution [15,22]. Although this is not a simple comparison of accuracy because the results have not been compared under the conditions that optimize the performance of the two methods, it should be considered that there is a need to set measurement conditions for MediaPipe as well and that data processing after the measurement alone is not sufficient. This problem may be solved using a machine learning method such as the recurrent neural network, which is also used for processing time series data, and by incorporating an algorithm that preserves information on the time axis before and after measurement.

One limitation of Fahrenheit is the large number of dropped frames. The breakdown of the subjects for whom falsely estimated frames were detected was five patients with BRS I and II, two patients with BRS III, one patient with BRS V, and one patient with BRS VI. The low body surface temperature of the subjects may have influenced the high number of dropped frames detected at lower motion speeds. The infrared sensor also provides quantitative information by measuring the difference in temperature between the fingers and the back of the hand. Patients with BRS I and II who have difficulty with voluntary movements tend to have weaker muscle contractions and lower body surface temperatures [23]. Limitations of infrared sensors in patients with severe peripheral edema have also been reported [24]. The detectable range of LMC is −30 °C to 60 °C [25], but it is possible that the difference in body surface temperature affected the accuracy of detection. In addition, stroke patients without joint motion are prone to peripheral edema in the acute phase [26], which may have also affected the mis-estimation. In other words, MediaPipe could not completely eliminate the noise due to a drop in accuracy, and Fahrenheit showed many false estimations in patients with BRS I and II, which may have affected the variability of data in patients with BRS I and II (DTW = 0.106–0.535, CCC = 0.251–0.963).

Fahrenheit operates at a frame rate of 60 fps, while MediaPipe processes at 30 fps. This disparity in frame rate can result in variations in dynamic motion measurements. To mitigate this, we downsampled the Fahrenheit data to 30 fps for a more accurate comparison. However, despite this adjustment, some variability may still exist, potentially impacting the comparability of the methods.

The exclusion criteria were established to ensure the reliability and accuracy of the measurements. However, we acknowledge that these criteria may limit the applicability of our findings to a broader patient population. Future studies could consider including a more diverse cohort to evaluate the generalizability of these results.

The primary distinction between Fahrenheit and MediaPipe systems lies in the number of joint landmarks tracked. Fahrenheit includes metacarpals of each finger, whereas MediaPipe does not, affecting the accuracy of MP joint flexion angle measurements. This limitation in MediaPipe’s joint landmarks may impact overall accuracy and clinical applicability. To address the constraint of joint landmarks in MediaPipe, particularly the absence of metacarpals which affects MP joint flexion angle precision, the following method is proposed: (1) Enhanced feature extraction, utilizing Feature Engineering + Long Short-Term Memory (FE + LSTM) models to derive more relevant features from existing landmarks. This approach can infer missing metacarpal positions and movements by enhancing data with temporal and spatial contextual information [27,28]. (2) Integration with additional sensors, combining MediaPipe with depth sensors or infrared cameras to enhance joint tracking capability [29]. (3) Machine learning models, incorporating human pose estimation models can help in capturing and predicting the missing joint movements by analyzing the seasonal and trend components of the hand movements [30,31]. This approach can model the temporal dependencies and seasonal variations in joint movements, thereby improving the accuracy of joint position estimations. (4) Algorithmic enhancements, implementing recurrent neural networks (RNNs) and LSTM networks trained on comprehensive datasets to predict missing joint positions [32]. These models effectively handle temporal dependencies and improve the robustness of motion tracking systems. By integrating STF + LSTM and FE + LSTM into future versions of our application, we can achieve more precise motion estimation and potentially overcome the current limitations of MediaPipe’s joint landmark detection.

## 5. Conclusions

The results showed a moderate-to-high correlation, indicating the possibility of transferring the functionality of Fahrenheit to MediaPipe, which can be used on smartphones and tablets. Fahrenheit’s weakness, which causes instability in the joint center estimation when a hand grasps an object (occlusion problem), can be solved by image processing using AI. MediaPipe has an Objectron library that can detect objects [12]. By developing an application that can run the MediaPipe hands and MediaPipe Objectron, it is possible to obtain the positional information of the object and hand simultaneously. This enables an accurate evaluation of human tool manipulation based on the positional relationship with an object, even when the hand is occluded. Currently, conventional motion captures only tracks human motion; the development of an application that can simultaneously track the motion of a human and that of an object generated by human motion can revolutionize the world of motion capture. The next task of our research team is to develop such an application and verify its performance in tracking hand motion with an object.

## Figures and Tables

**Figure 1 biomimetics-09-00400-f001:**
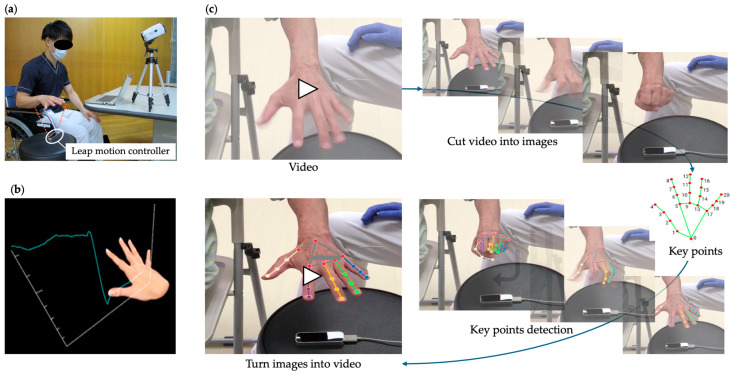
Fahrenheit and MediaPipe measurement method. (**a**) Patient position during measurement. (**b**) Virtual hand motion displayed in real time. The vertical axis indicates the numerical value of finger movement, while the horizontal axis indicates the time of measurement. (**c**) Keypoint detection method of MediaPipe for video data. The video is cut out at 30 frames per second, the keypoints are detected, and the successive images are merged into a video.

**Figure 2 biomimetics-09-00400-f002:**
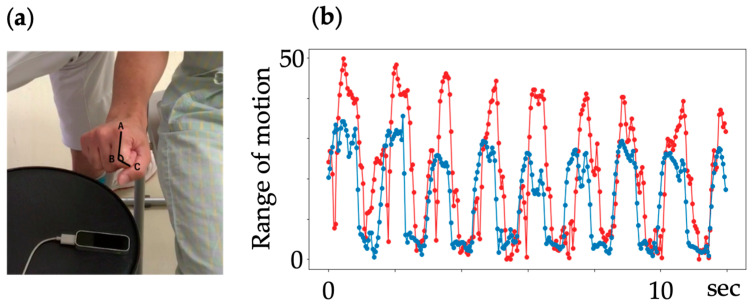
Method of angle conversion. (**a**) Calculation method for joint angles. (**b**) An example of the resulting motion trajectory. The motion trajectories of Fahrenheit and MediaPipe are shown as blue and red lines in the plots, respectively. The horizontal and vertical axes represent time and joint angle, respectively. (**a**) Demonstrates that the maximum achievable range of motion for the patient is realistic, and the flexion of the finger joint appears to be close to 90 degrees. (**b**) Illustrates the range of motion measured during the experiment, which is less than 90 degrees for Fahrenheit and even smaller angles for MediaPipe. These results may differ from the actual angles observed in the captured images because Fahrenheit reconstructs images from two cameras, while MediaPipe estimates angles from two-dimensional images.

**Figure 3 biomimetics-09-00400-f003:**
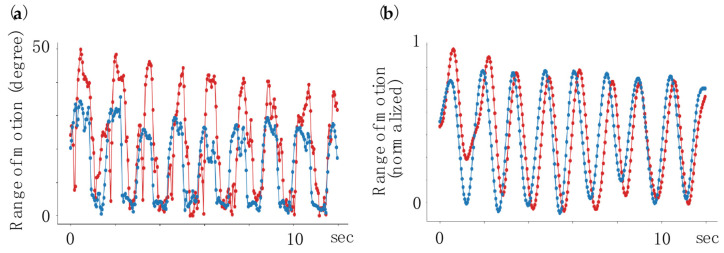
Smoothing of kinematics data obtained by MediaPipe and Fahrenheit. (**a**) Raw data. (**b**) Data after smoothing. Blue and red plots indicate the motion trajectories of Fahrenheit and MediaPipe, respectively.

**Figure 4 biomimetics-09-00400-f004:**
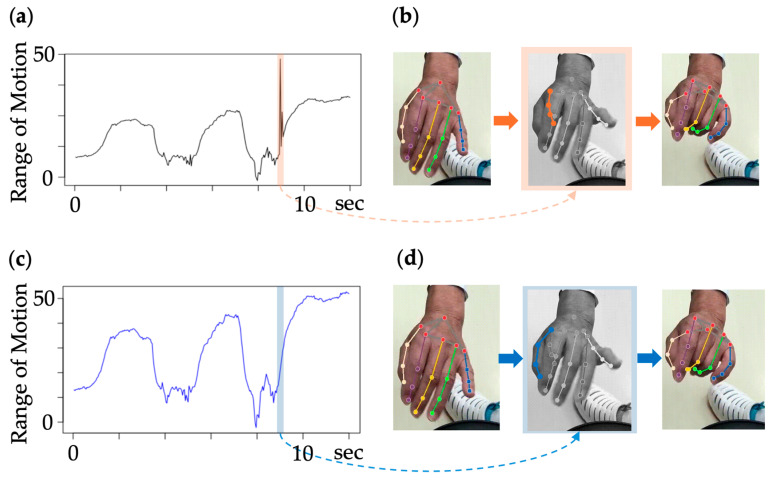
Detection and completion of misestimated frames and completion method. (**a**) The absolute difference dx in the mother finger PIP. Values exceeding the mean ± 3SD are detected in the area circled in red. (**b**) The time frames. (**c**) Implementation of spline smoothing, excluding the misestimated frames. (**d**) Excluded frames.

**Figure 5 biomimetics-09-00400-f005:**
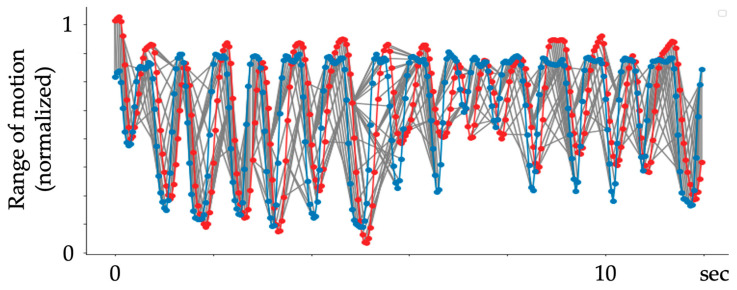
Dynamic Time Warping (DTW) distance calculation. Blue and red indicate the motion trajectories of Fahrenheit and Mediapipe, respectively. The line connecting them indicates the DTW distance.

**Figure 6 biomimetics-09-00400-f006:**
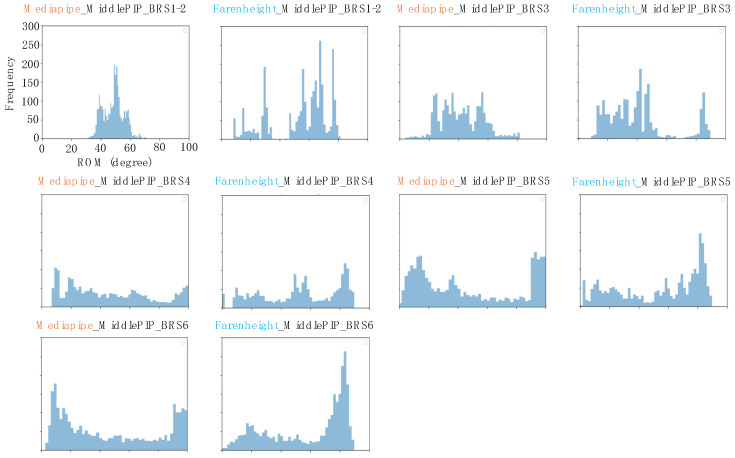
Histograms for each Brunnstrom Recovery Stage (BRS) phase. The *x*-axis indicates the observed range of motion (ROM), divided into intervals of 2° each. The *y*-axis indicates the number of data observed in the interval. Each histogram includes the data of all the subjects divided into BRS phases.

**Figure 7 biomimetics-09-00400-f007:**
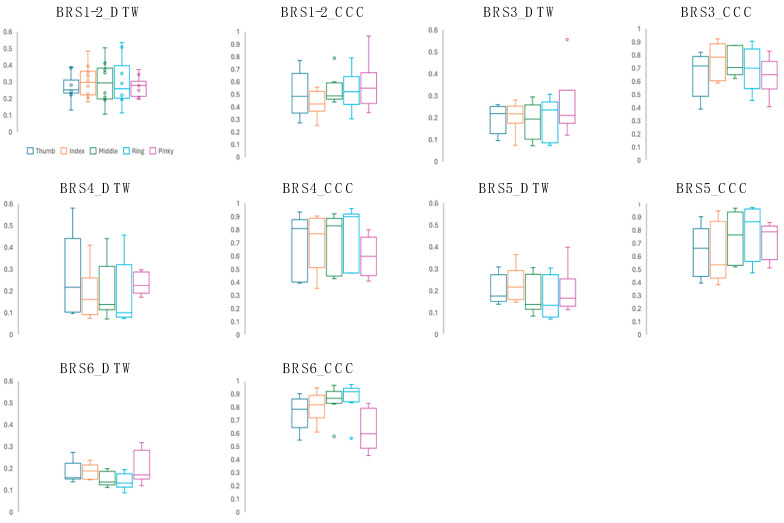
Comparison of the agreement of the measurements. Dynamic Time Warping (DTW) distance between MediaPipe and Fahrenheit for each subject is shown on the left and cross-correlation analysis is shown on the right for each Brunnstrom Recovery Stage (BRS). The bottom line of the error bar indicates the minimum value, the center line indicates the median value, the × mark indicates the mean value, and the top line indicates the maximum value. Boxes indicate interquartile ranges.

**Table 1 biomimetics-09-00400-t001:** Comparison of basic performance.

	MediaPipe	Fahrenheit
Frame rate	30	60 * detune to 30
Number of joint landmarks	21	24
Observable joint range of motion	180	90

Values are expressed as real numbers. * Fahrenheit was downsampled and analyzed at 30 frames per second (fps).

**Table 2 biomimetics-09-00400-t002:** Demographic data of the subjects.

Attribute	Classification	Values
Age		71 ± 11
Gender	Male/Female	30/10
Paralyzed side	Right/Left	23/17
Day of a patient’s illness		13.5 ± 14.8
Diagnosis	Cerebral hemorrhage/cerebral infarction	9/31
Brunnstrom stage (finger)	I/II/III/IV/V/VI	2/8/6/6/9/9

Categorical data are presented as real values, and numerical data are presented as mean ± standard deviation.

**Table 3 biomimetics-09-00400-t003:** Comparison of corrections.

	MediaPipe	Fahrenheit
Subjects with mis-estimated frames detected	3/40	9/40
MAD of raw data	2.46 ± 0.87	1.42 ± 0.46
MAD after preprocessing	0.81 ± 0.32	0.02 ± 0.01

Categorical data are presented as real values, and numerical data are presented as mean ± standard deviation.

**Table 4 biomimetics-09-00400-t004:** Maximum flexion angle.

Peak_Flexion (deg)	BRS I–II (*n* = 10)	BRS III (*n* = 6)	BRS IV (*n* = 6)	BRS V (*n* = 9)	BRS VI (*n* = 9)
MediaPipe					
Thumb	35.1 ± 14.4	34.9 ± 14.5	55.6 ± 16.3	41.3 ± 12.6	42.3 ± 8.1
Index	54.5 ± 15.6	45.5 ± 26.6	75.9 ± 25.9	93.5 ± 10.7	99.2 ± 0.8
Middle	52.8 ± 9.4	56.0 ± 20.6	87.6 ± 11.3	99.5 ± 0.6	98.6 ± 3.2
Ring	60.8 ± 8.8	60.7 ± 18.2	91.5 ± 7.2	99.5 ± 1.1	99.4 ± 1.1
Pinky	52.8 ± 10.2	59.9 ± 21.2	68.5 ± 15.7	98.9 ± 1.3	97.4 ± 6.0
Fahrenheit					
Thumb	53.3 ± 24.0	52.1 ± 22.0	68.3 ± 17.7	31.5 ± 6.0	40.2 ± 15.4
Index	60.1 ± 19.4	52.9 ± 22.4	73.6 ± 20.0	86.4 ± 3.1	89.1 ± 1.3
Middle	60.5 ± 18.1	51.6 ± 20.6	73.3 ± 20.9	87.1 ± 2.6	88.1 ± 2.2
Ring	59.7 ± 17.7	51.3 ± 21.0	72.4 ± 19.7	86.8 ± 3.1	88.0 ± 2.5
Pinky	18.2 ± 11.6	14.2 ± 22.2	37.8 ± 15.7	87.2 ± 2.3	88.1 ± 2.7

Values are presented as mean ± standard deviation.

**Table 5 biomimetics-09-00400-t005:** Comparison of angular velocities.

Peak_Velocity/Average_Velocity (deg/s)	BRS I–II (*n* = 10)	BRS III (*n* = 6)	BRS IV (*n* = 6)	BRS V (*n* = 9)	BRS VI (*n* = 9)
MediaPipe					
Thumb	87.9/6.2	111.2/14.7	274.7/55.3	667.6/110.1	780.8/186.9
Index	82.6/9.8	236.1/29.7	505.0/101.6	1296.6/231.1	1400.5/365.9
Middle	61.1/9.7	240.0/26.8	544.8/108.6	1309.0/239.9	1425.9/380.4
Ring	58.5/9.4	256.2/27.6	573.6/110.6	1233.8/227.6	1392.0/365.5
Pinky	104.2/9.6	187.8/25.8	574.8/96.6	1198.8/177.8	1231.8/282.3
Fahrenheit					
Thumb	43.7/7.9	126.0/10.8	167.9/49.6	271.7/99.4	672.0/207.5
Index	72.7/12.3	214.0/28.3	506.4/114.2	1295.8/214.7	1412.8/355.7
Middle	69.2/13.2	225.8/27.0	501.6/102.6	1240.1/217.7	1448.2/373.8
Ring	77.5/14.9	216.8/24.7	587.2/100.2	1300.6/200.2	1403.1/351.1
Pinky	79.9/13.3	165.7/22.5	501.6/74.4	1225.1/205.3	1208.5/345.7

Each cell shows the maximum or average angular velocity. Values are shown as mean ± standard deviation.

## Data Availability

Study data will be provided upon request from the corresponding authors.

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
