# Peer review of "Verification of Criterion-Related Validity for Developing a Markerless Hand Tracking Device"

_biomimetics, 2024, doi:10.3390/biomimetics9070400_

Round 1

Reviewer 1 Report

Comments and Suggestions for Authors

The article presents an evaluation of a method for assessing hand motor function in stroke patients using a smartphone application integrated with MediaPipe. The authors aim to demonstrate the criterion-related validity of MediaPipe by comparing it to Fahrenheit, a pre-existing device that uses infrared camera image processing.

The research addresses an important need in rehabilitation by providing a potentially accessible tool for assessing hand function. However, there are both strengths and weaknesses in the article's presentation, methodology, and results.

Strengths:

1. Using MediaPipe for hand tracking.

2. The article states its objective to measure hand movements in stroke patients using both MediaPipe and Fahrenheit and to assess their criterion-related validity.

3. The inclusion of patients and the retrospective observational design are appropriate for the aims of the research. The use of the Brunnstrom Recovery Stages (BRS) to classify the severity of paralysis adds robustness to the analysis.

4. The description of the measurement of hand movements protocol.

5. The use of G*power to determine sample size and statistical methods to validate the results.

Weaknesses:

1. While the comparison between Fahrenheit and MediaPipe is a core part of the research, the limited discussion of differences in joint landmarks (e.g., the absence of metacarpals in MediaPipe) and their potential impact on results could be expanded. Add description of other state-of-the-art work related to gesture recognition and MediaPipe. See SOTA AUTSL (https://paperswithcode.com/sota/sign-language-recognition-on-autsl) - STF+LSTM and FE+LSTM.

2. The exclusion criteria are fairly stringent, which may limit the applicability of the results to a broader patient population.

3. The inherent differences in frame rates between MediaPipe and Fahrenheit (30 fps vs. 60 fps) may introduce variability that could affect the equivalence of the methods.

Comments on the Quality of English Language

Moderate editing of English language required.

Reviewer 2 Report

Comments and Suggestions for Authors

The authors developed an algorithm to estimate hand paralysis, and in this paper, they propose a developed smartphone app that integrates a new algorithm. The objective was to measure hand movements in stroke patients using both algorithms. The topic is very interesting, and the paper is well-written. However, to improve it a little bit, I recommend:

  1. In figure 1 (c), it is small and difficult to see.
  2. In figure 2 (a), the angle represented is more than 90 degrees, and in 2 (b), the ROM is less than 90 degrees. What is the reason for this?
  3. In figure 4 (b) and (d), it is difficult to see the four key points for the thumb.

Round 2

Reviewer 1 Report

Comments and Suggestions for Authors

The authors have not corrected shortcoming 1, namely:

1. While the comparison between Fahrenheit and MediaPipe is a core part of the research, the limited discussion of differences in joint landmarks (e.g., the absence of metacarpals in MediaPipe) and their potential impact on results could be expanded. Add description of other state-of-the-art work related to gesture recognition and MediaPipe. See SOTA AUTSL (https://paperswithcode.com/sota/sign-language-recognition-on-autsl) - STF+LSTM and FE+LSTM.

This is a research article, and you should not just add a paperwithcode link, but analyze the best articles from the list.

Comments on the Quality of English Language

Minor editing of English language required.
